# Hyperprolactinemia in Adults with Prader-Willi Syndrome

**DOI:** 10.3390/jcm10163613

**Published:** 2021-08-16

**Authors:** Anna Sjöström, Karlijn Pellikaan, Henrik Sjöström, Anthony P. Goldstone, Graziano Grugni, Antonino Crinò, Laura C. G. De Graaff, Charlotte Höybye

**Affiliations:** 1Department of Clinical Chemistry, Karolinska University Hospital, 171 76 Stockholm, Sweden; anna.sjostrom@sll.se; 2Department of Molecular Medicine and Surgery, Karolinska Institutet, 171 76 Stockholm, Sweden; 3Internal Medicine, Division of Endocrinology, Erasmus MC, University Medical Centre Rotterdam, 3000 CA Rotterdam, The Netherlands; k.pellikaan@erasmusmc.nl (K.P.); l.degraaff@erasmusmc.nl (L.C.G.D.G.); 4Internal Medicine, Center for Adults with Rare Genetic Syndromes, Division of Endocrinology, Erasmus MC, University Medical Center Rotterdam, 3000 CA Rotterdam, The Netherlands; 5Dutch Center of Reference for Prader-Willi Syndrome, 3015 GD Rotterdam, The Netherlands; 6Academic Centre for Growth Disorders, Erasmus MC, University Medical Centre Rotterdam, 3000 CA Rotterdam, The Netherlands; 7Department of Clinical Neuroscience, Karolinska Institutet, 171 76 Stockholm, Sweden; henrik.sjostrom@ki.se; 8Center for Neurology, Academic Specialist Center, 113 65 Stockholm, Sweden; 9PsychoNeuroEndocrinology Research Group, Centre for Neuropsychopharmacology, Division of Psychiatry, Department of Brain Sciences, Faculty of Medicine, Imperial College London, Hammersmith Hospital, London W12 0NN, UK; tony.goldstone@imperial.ac.uk; 10Department of Endocrinology, Imperial College Healthcare NHS Trust, Hammersmith Hospital, London W12 0HS, UK; 11International Network for Research, Management and Education on Adults with PWS; g.grugni@auxologico.it (G.G.); nino3381@gmail.com (A.C.); 12Divison of Auxology, Istituto Auxologico Italiano, IRCCS, 28824 Piancavallo, Italy; 13European Reference Network on Rare Endocrine Conditions; 14Reference Center for Prader-Willi Syndrome, Bambino Gesù Hospital, Research Institute, Palidoro, 00050 Rome, Italy; 15Department of Endocrinology, Karolinska University Hospital, 171 76 Stockholm, Sweden

**Keywords:** Prader-Willi syndrome, adults, hyperprolactinemia, hypogonadism

## Abstract

Prader-Willi syndrome (PWS) is a rare neurodevelopmental genetic disorder typically characterized by body composition abnormalities, hyperphagia, behavioural challenges, cognitive dysfunction, and hypogonadism. Psychotic illness is common, particularly in patients with maternal uniparental disomy (mUPD), and antipsychotic medications can result in hyperprolactinemia. Information about hyperprolactinemia and its potential clinical consequences in PWS is sparse. Here, we present data from an international, observational study of 45 adults with PWS and hyperprolactinemia. Estimated prevalence of hyperprolactinemia in a subset of centres with available data was 22%, with 66% of those related to medication and 55% due to antipsychotics. Thirty-three patients were men, 12 women. Median age was 29 years, median BMI 29.8 kg/m^2^, 13 had mUPD. Median prolactin was 680 mIU/L (range 329–5702). Prolactin levels were higher in women and patients with mUPD, with only 3 patients having severe hyperprolactinemia. Thyroid function tests were normal, 24 were treated with growth hormone, 29 with sex steroids, and 20 with antipsychotic medications. One patient had kidney insufficiency, and one a microprolactinoma. In conclusion, severe hyperprolactinemia was rare, and the most common aetiology of hyperprolactinemia was treatment with antipsychotic medications. Although significant clinical consequences could not be determined, potential negative long-term effects of moderate or severe hyperprolactinemia cannot be excluded. Our results suggest including measurements of prolactin in the follow-up of adults with PWS, especially in those on treatment with antipsychotics.

## 1. Introduction

Prader-Willi syndrome (PWS) is a rare and complex neurodevelopmental disorder, characterized by hypothalamic dysfunction. PWS is caused by a lack of expression of paternally inherited genes in the PWS region of chromosome 15q11–13 [1]. Approximately 65–70% of the patients have a paternal deletion, 30% a maternal uniparental disomy (mUPD), 2–5% an imprinting defect, and 0.1% chromosomal translocation [1,2]. In adults, PWS is clinically characterized by endocrine deficiencies, hyperphagia, obesity and its associated comorbidities, intellectual disability, and a characteristic neuropsychological and behavioural profile [1,2]. The incidence of psychosis in adults with PWS is high (10–20%), particularly in those with the mUPD genotype [1,2]. Hypogonadism with various degrees of genital hypoplasia, delayed or incomplete pubertal development, and infertility is seen in about 90% and involves both hypothalamic and primary gonadal abnormalities [1,2,3,4,5,6]. PWS is associated with hypogonadism irrespective of prolactin levels. 

Prolactin is produced by the lactotroph cells in the anterior gland of the pituitary [7,8]. Dopamine from the hypothalamus exerts a tonic inhibition on pituitary prolactin production and secretion [7,8]. Any condition, hypothalamic disorder or medication interfering with dopamine secretion or action (e.g., antipsychotics, some antidepressants) might lead to an increase in prolactin levels, i.e., hyperprolactinemia [7,8,9]. Prolactin can also be released from the hypothalamic paraventricular nucleus and medial pre-optic area, in response to physiological stimuli, including stress [7,8]. The main physiological effects of prolactin are enlargement of breasts during pregnancy, milk production, amenorrhea during breast feeding, and stimulation of immune system-growth factors [7,8]. 

Hyperprolactinemia inhibits the pulsatile secretion of gonadotropin releasing hormone (GnRH), and thereby the production of gonadotropins (FSH and LH), which results in a reduction of sex hormones and development of hypogonadism [7,8,10,11]. Hypogonadism can cause decreased libido, oligo- or amenorrhea and infertility in women, and decreased libido, impotence, infertility, and gynecomastia in men [7,8,10,11]. Galactorrhoea might be present in both genders, but is more prevalent in women. Osteoporosis is common among patients with hyperprolactinemia and is considered to be the result of hyperprolactinemia-induced hypogonadism [11]. Furthermore, severe hyperprolactinemia can lead to unfavourable metabolic effects, including increased blood glucose, LDL-cholesterol and triglycerides levels and has been shown to increase cardiovascular mortality in non-PWS males [7,8,12,13]. Treatment with dopamine agonists such as cabergoline can reverse these effects as well as decrease body mass index (BMI) and total body fat in patients with hyperprolactinemia [13]. 

Among the adverse effects of hyperprolactinemia, it is of particular importance in PWS to consider the potential unfavourable metabolic effects due to the hyperphagia, a decreased basal metabolic rate and high risk of developing morbid obesity [2,14]. Several causes of hyperprolactinemia might be present in PWS. For example, the patients are easily stressed when routines are changed, and many patients have psychotic illnesses, depression, or mood instability treated with antipsychotics (dopamine antagonists) or antidepressants (tricyclics or selective serotonin reuptake inhibitors, SSRIs) that can cause hyperprolactinaemia [9,14]. To date, high levels of prolactin in PWS adults have been reported anecdotally in both sexes, especially during therapy with psychotropic medications, and a thorough assessment of hyperprolactinemia in PWS is lacking. The knowledge about prolactin levels in PWS is sparse, and the aim of this study was to characterize adults with PWS and hyperprolactinemia.

## 2. Patients and Methods

Data was collected from adults with PWS from reference centres for PWS in Rotterdam (Netherlands), Rome and Piancavallo (Italy), London (UK) and Stockholm (Sweden). Demographic data, BMI, co-morbidities, medication, and results of blood tests were collated from the patients’ medical records. 

All clinic visits and assessments were part of routine clinical practice. BMI was calculated as weight divided by the square of height in meters, kg/m^2^. BMI from 18.5–25 kg/m^2^ was defined as normal, between 25–30 kg/m^2^ as overweight and above 30 kg/m^2^ as obese, according to the WHO criteria. Results for analyses of prolactin, sex hormones, gonadotropins, TSH, free thyroxine (fT4), free triiodothyronine (fT3), kidney, and liver function were retrieved from the patients’ medical records. Analyses were performed according to local assays. The same methodology for analysis was used in the participating sites, but with different equipment, see appendix Table A1. All prolactin concentrations and reference values were converted to mIU/L. The overall reference range for prolactin in men was 0–500 mIU/L between centres (lower reference limit between 0 and 100; upper reference limit between 300 and 500) and for women 85–700 mIU/L (lower reference limit between 85 and 102; upper reference limit between 490 and 700). 

In the individual centres, it was also noted if prolactin concentrations had been measured on just a single occasion, if macroprolactin (an immunological artefact of no known clinical significance that raises measured prolactin concentrations [15]) was analysed, and if performed, the results of MRI or CT imaging of the pituitary to exclude a visible pituitary adenoma.

This study was approved by the ethical committee of participating centres and/or informed consent was collected from individual patients in accordance with national laws and regulations. 

### Statistics 

Data are presented as median (range). Statistical analysis was performed using SPSS (version 26.0 for Mac, IBM Corp., Armonk, NY, USA). Differences between groups were calculated by Mann–Whitney U-test. A two-way ANCOVA was conducted to examine differences in prolactin levels depending on gender and the two most common genetic factors in the study; paternal deletion and mUPD. In this analysis, age was entered as a covariate. Statistical significance was set at *p* < 0.05.

## 3. Results

A total of 441 adults with PWS were seen in the participating clinics. Among them 45 patients (33 men and 12 women) (15 from the Netherlands, seven from Italy, ten from United Kingdom and 13 from Sweden) had hyperprolactinemia. The hyperprolactinemia was diagnosed both without clinical suspicion, and by routine measurements. Repeated prolactin measurements (more than once) were performed in 37 patients, in whom prolactin remained elevated, while in eight patients the prolactin was measured only on one occasion. 

It was not possible to determine the true prevalence of hyperprolactinemia in the whole cohort, since not all patients had routine measurements of prolactin or had data available on the number of patients with a non-elevated prolactin. However, in those centres where all prolactin results were available, and measurement was part of a routine assessment, the prevalence of hyperprolactinemia was as follows: Netherlands, 15 out of 89 (16.9%) with 73% of those related to medication (62% of total with hyperprolactinemia on antipsychotics), UK, 10 out of 50 (20.0%), with 50% of those related to medication (40% antipsychotics), and Sweden, 13 out of 33 (39.4%) with 69% of those related to medication (62% antipsychotics), giving an average prevalence of 38 out of 172 (22.1%), with 66% of those related to medication (55% antipsychotics).

The patients with hyperprolactinemia had a median age of 29 years (range 19–58) and median BMI was 29.8 kg/m^2^ (range 20.5–45.5) (Table 1). Twenty-four patients had a paternal deletion (53.3%), 13 mUPD (31.1%) and three an imprinting defect or chromosomal translocation (6.7%). Four were *SNRPN* methylation positive for PWS, but further genetic characterization was not available (8.9%). The distribution of patients from each site were similar regarding gender and genotypes.

Median prolactin was 680 mIU/L (range 329–5950), 615 mIU/L in men and 1061 mIU/L in women (Figure 1), and 625 mIU/L (range 360–1190) in patients with paternal deletion and 695 mIU/L (range 337–1421) in patients with mUPD (Figure 2). One man and two women had prolactin levels >2200 mIU/L. Using two-way ANCOVA to assess the patients with paternal deletion and mUPD, we found higher prolactin levels in patients with mUPD compared to those with paternal deletion (*p* = 0.005), and higher prolactin in women compared to men (*p* = 0.003). For the two-way ANCOVA analysis, two outliers with paternal deletion were excluded; one with a known microprolactinoma and one being the only patient from a site and suspected to be due to a different laboratory methodology. 

Macroprolactin was not found in any of the 17 patients in whom it was looked for (nine out of 10 patients in UK, seven out of seven patients in Italy and one patient out of 15 patients in The Netherlands). 

Median serum TSH was 1.6 mU/L (range 0.3–3.8), fT4 15 pmol/L (range 8.2–22.2) and fT3 4.4 pmol/L (range 2.9–5.4). Several different assay methods were used, but thyroid function was determined as normal in all patients; in particular, none of the patients had primary hypothyroidism with TSH above the local reference value. Two patients (4.4%) were treated with levothyroxine. Twenty-four patients (53.3%), 17 men and seven women, were treated with growth hormones. One patient had renal insufficiency (2.2%), while none of the patients had liver insufficiency. Four patients had type 2 diabetes mellitus (8.9%). Twenty-two men and seven women were treated with sex steroids for hypogonadism. Sixteen other patients had untreated hypogonadism.

For most of the patients, behavioural or psychiatric problems were reported. In total, 28/45 (62%) were on a medication known to cause hyperprolactinemia. Among them, 14 men and four women were treated with risperidone, two men with methoprazine, three men with quetiapine, one man with piamperone, one man with olanzapine, and one woman with levopromazin, totalling 25) on an antipsychotic medication (56% of all patients with raised prolactin), while 9 (20%) were on an SSRI (sertraline, fluoxetine or citalopram), of whom three were also on an antipsychotic medication. 

Only twelve patients (26.7%) underwent radiological examinations (11 MRI, one CT) of the pituitary or brain. Amongst this one man had a cystic microprolactinoma with a prolactin of 5702 mIU/L at diagnosis. The hyperprolactinemia normalised on treatment with cabergoline. This patient and a woman with hyperprolactinemia were the only patients who received dopamine agonist treatment. In the UK cohort, 8 out of 10 of the patients with hyperprolactinaemia had a dedicated MRI pituitary scan with gadolinium contrast of which all were normal. In these UK patients, median (range) prolactin was 617 (386–3488) mIU/L, and 62.5% were on psychotropic medication(s) that may raise prolactin (antipsychotic or antidepressant). In the Swedish cohort, one patient underwent a CT scan of the brain and one patient MRI of the brain, but these examinations were not dedicated for the pituitary. 

The patient with renal insufficiency underwent kidney transplantation the year before this study. The patient was not treated with antipsychotic medication, but was on a stable dosage of testosterone replacement therapy. Over time prolactin levels gradually increased as the kidney function decreased. Two years before the kidney transplantation eGFR was 11 mL/min (reference value > 80) and prolactin 787 mIU/L. After the transplantation eGFR increased to 58 mL/min and prolactin decreased to 175 mIU/L.

No clear clinical effects of hyperprolactinemia were identified in our cohort. Galactorrhoea or gynecomastia were not reported in any patient. 

## 4. Discussion

In this study of 45 adults with PWS and hyperprolactinemia the majority were men, and the most frequent aetiology was treatment with antipsychotic medication in 56%. The prolactin levels were generally only mildly increased and were higher in women and in patients with mUPD. Clear clinical consequences of the hyperprolactinemia were not observed. 

Hyperprolactinemia is defined as sustained levels of prolactin above the laboratory upper limit of normal [7,8]. A grading of hyperprolactinemia in psychotic patients according to its clinical severity has been suggested [16]. Accordingly, hyperprolactinemia can be mild (<1100 mIU/L or <50 ng/mL), moderate (1100–1600 mIU/L or 51–75 ng/mL), or severe (>2200 mIU/L or >100 ng/mL). Prolactin levels between 1600 mIU/L or 75 ng/L and 2200 mIU/L or 100 ng/L were not included in the grading, but in the present study they were considered as moderately increased. Using these definitions, the hyperprolactinemia in the present study was generally mild (80.0%), with only six patients (13.3%) having moderate and three (6.7%) severe hyperprolactinemia. Hyperprolactinemia can cause hypogonadism, reduced bone mineralization and an increased cardiovascular risk [7,8,11,12,13,14,17,18]. In PWS, these risks are already high and mainly related to hypogonadism, GH deficiency and obesity, independent of hyperprolactinemia. In eugonadal patients or patients with mild hypogonadism, hyperprolactinemia might further lower sex hormone levels further increasing the risks. Clinical consequences of hyperprolactinemia were not observed in the present study, probably because prolactin was only mildly increased. 

Several factors can affect prolactin measurements, and to minimize the confounding effect of them, blood samples should be taken in an un-stressed condition [19,20]. In patients with PWS, blood sampling is often stressful and difficult due to small, thin veins. Presuming the stress elicited by taking blood samples decreases when the procedure is repeated, this was not the case in 37 patients in whom hyperprolactinemia continued to be elevated in repeated measurements, indicating that the stress did not decrease, or the hyperprolactinemia had another aetiology. Other than medications as discussed below, this could be related to hypothalamic defects in PWS interrupting the dopaminergic inhibition of pituitary prolactin secretion.

Moreover, the release of prolactin is pulsatile with the highest levels in the early morning and measurement of prolactin is recommended 2–3 h after awakening [19,20]. The presence of macroprolactin might falsely increase the prolactin level [19,20,21,22]. Most circulating prolactin is monomeric (23 kDa), but prolactin also circulates in larger isoforms called macroprolactin. These are complexes between prolactin and IgG antibodies causing hyperprolactinemia through reduced clearance and are not considered to have any clinical effect [15,19,20,21,22]. In the present study, the presence of macroprolactin was not controlled for and excluded in all laboratories’ prolactin analyses. However, in our view, the presence of macroprolactin was unlikely to have complicated our results as in the 37 patients in whom it was measured with negative macroprolactin. 

The aetiology of hyperprolactinemia can be physiological like in pregnancy and breast feeding, stress, exercise, or food intake [7,8]. It can also be pharmacological, induced by several drugs (antipsychotics, antidepressants, opioids, phenytoin, verapamil, antiemetic, oestrogen, metoclopramide, cimetidine, omeprazole, and several other medications) [7,8,9]. Other aetiologies are hypothalamic tumours, prolactinomas or other pituitary tumours and after radiotherapy of the area [7,8]. Furthermore, primary hypothyroidism, chronic renal insufficiency, liver cirrhosis and polycystic ovarian syndrome (PCOS) can lead to hyperprolactinemia and finally hyperprolactinemia can be idiopathic [7,8]. It is, therefore, important to consider secondary causes through a careful medical history including use of drugs, clinical examination, and evaluation of kidney, liver and thyroid gland function and exclusion of pregnancy in fertile women [7,8]. In our study, the most common aetiology of hyperprolactinemia was the treatment with antipsychotic medication. Psychosis is more common in patients with mUPD [2,23] and approximately 30% of our cohort had mUPD. In our cohort, prolactin levels were higher in patients with mUPD compared to patients with paternal deletion. The reason for this could be that patients with mUPD were more frequently treated with antipsychotics. Only one patient had a microprolactinoma. One patient had previously suffered from severe renal insufficiency, and during that time, prolactin increased. After kidney transplantation and improvement in kidney function, prolactin normalized. 

Hyperprolactinemia is a common consequence of treatment with some antipsychotics, mainly risperidone, paliperidone, and amisulpride, but less frequently related to olanzapine [24]. Dopamine binds to D2 receptors on the lactotroph cells in the pituitary, which inhibits both the synthesis and secretion of prolactin [24]. Administration of dopamine antagonists, such as antipsychotics, might therefore lead to hyperprolactinemia. The highest rates of hyperprolactinemia are consistently reported for conventional antipsychotic drugs, but hyperprolactinemia also occurs with the atypical antipsychotics, but with aripiprazole and quetiapine having the most favourable profile [24]. There are large variations in the individual increase in prolactin caused by antipsychotic drugs and transient elevations can also be seen [24]. In a recent study of 170 patients treated with antipsychotics, female gender was associated with an increase in serum prolactin levels [25]. In the present study, risperidone was the most frequently used antipsychotic medication. Too few patients were on other antipsychotics for a difference in prolactin levels to be assessed. 

It was difficult to determine the overall prevalence of hyperprolactinemia in all centres, but in individual centres where sufficient data was available, the prevalence varied from 20–39%, averaging 22% with 66% of those related to medication (55% antipsychotics). Since this proportion of these patients who were on medications elevating prolactin was similar to that in the whole cohort, it is likely that this prevalence is a reasonable estimate of true prevalence in the patients seen in all these centres. However, it should be noted that there may be a referral bias for patients with behavioural issues, and therefore on psychotropic medication to these referral centres. In comparison, cross-sectional studies of patients with schizophrenia and bipolar diseases have reported a prevalence of hyperprolactinemia of 44–75% in women and 23–72% in men [26]. Considering, that 80% of our PWS patients with hyperprolactinemia had only a mild hyperprolactinemia and that some antipsychotics (such as risperidone, paliperidone and amisulpride) might raise the serum prolactin level at relatively low doses [24], we have no indications that patients with PWS have a greater susceptibility for developing hyperprolactinemia during anti-psychotic treatment than other patient groups.

Studies of non-PWS adults with psychotic illnesses treated with antipsychotics have reported a frequency of galactorrhoea of 10–90% in women, whereas it is rarely reported in men [14,15,27]. It has also been shown that gynecomastia occurs in 1–11% of men [27]. However, galactorrhoea and gynecomastia were not reported in any of the patients in the present study. Menstrual irregularities in non-PWS women with schizophrenia receiving long term treatment with conventional antipsychotics or risperidone were reported with a prevalence ranging between 25–78% [17,23]. Due to hypogonadism in PWS and the use of sex hormone replacement therapy, it was not possible to evaluate this issue in the current study. Long-term treatment with antipsychotic medications and hyperprolactinemia have been observed to decrease bone mineral density in 32–65% of patients on antipsychotics, but confounders like poor diet, low physical activity, smoking, low exposure to sun light were considered to affect the results [17,23]. Bone mineral density measurements were not available for the present cohort and the effect on bone mineral density is therefore unknown. 

An interesting aspect is whether hyperprolactinemia due to treatment with antipsychotic medications might induce a prolactin producing pituitary adenoma. Long-term treatment with antipsychotics in mice has led to the development of prolactinomas [28], but from currently available pharmacovigilance data it is difficult to conclude whether antipsychotics are implicated in the development or progression of prolactinomas [28,29,30]. Only one patient in the present cohort was diagnosed with a microprolactinoma and this patient did not receive treatment with antipsychotics. A microprolactinoma has been previously reported in only one patient with PWS [31]. However, only 12 patients in our cohort underwent radiological examination of the pituitary or brain. Given the lack of pituitary macroadenomas and rarity of visible microprolactinomas in our cohort, it remains to be determined under which criteria a dedicated MRI or CT scan of the pituitary gland should be considered if an adult with PWS is found to have a raised prolactin. If a clear reason for the hyperprolactinemia is lacking perhaps hyperprolactinemia in the severe range (>2200 mIU/L or >100 ng/mL) might be an appropriate cut-off to warrant dedicated pituitary imaging (6.7% of our cohort had a prolactin level in the severe range, including one patient with a microprolactinoma). Given the possibility of causing a pituitary adenoma, a dedicated MRI or CT of the pituitary should also be considered in patients with moderately increased prolactin levels for long-term.

When indicated hyperprolactinemia is usually treated with dopamine agonists. The most common side effects of dopamine agonists are nausea, vomiting, dizziness, postural hypotension, headache, nasal congestion, and constipation. In addition to these side effects, dopamine agonist treatment can trigger or worsen an impulse control disorder, such as hypersexuality and gambling addiction, or psychotic illness. Furthermore, long-term treatment with ergot-derived dopamine agonists (including carbergoline) in patients with Parkinson’s disease has shown an increased incidence of heart valve insufficiencies, though at much higher doses than usually used to treat hyperprolactinaemia from pituitary adenomas [32,33]. Concerning treatment with cabergoline in patients with prolactinomas, no increased risk of clinically significant valve insufficiency has been found to date [32,33]. Only two patients were treated with cabergoline and treatment was successful and without side effects. However, attention should always be paid to potential mental side effects of dopamine agonists in a vulnerable group of patients such as patients with PWS. 

The strength of the study is the careful evaluation of all patients with hyperprolactinemia and the use of the same method for analysis of prolactin in all countries. Limitations to the study are the retrospective character and that equipment from different manufacturers were used for the analysis of prolactin in each country. However, after conversion of all prolactin levels to mIU/L the differences in upper and lower reference values was not considered of significance for the results of this study, in contrast to the clinical settings where an exact value is of importance. Furthermore, not all countries had formally excluded macroprolactin, and in some of the patients the diagnosis of hyperprolactinemia was based on randomly measured prolactin as not all countries measured prolactin concentrations regularly in every patient. For this reason, it was not possible to determine the true prevalence of hyperprolactinemia in PWS.

## 5. Conclusions

The present cohort of 45 adults with PWS and hyperprolactinemia consisted of more men than women. Prolactin levels were higher in women and in patients with mUPD. Estimated prevalence was 22%, with two thirds of those related to psychotropic medication with the most common aetiology antipsychotic medication in 55%. In most cases, the hyperprolactinemia was mild and without clear clinical consequences. However, potential negative long-term effects cannot be excluded. Due to cognitive impairment, behavioural problems, and hypogonadism, the clinical effects of hyperprolactinemia might not be noticed. Therefore, it is important to routinely measure serum prolactin concentrations, especially during treatment with antipsychotics, to repeat measurements if an initial level is raised, to exclude macroprolactin artefact, and to consider a dedicated MRI or CT pituitary scan if there is moderate or severe hyperprolactinemia.

## Figures and Tables

**Figure 1 jcm-10-03613-f001:**
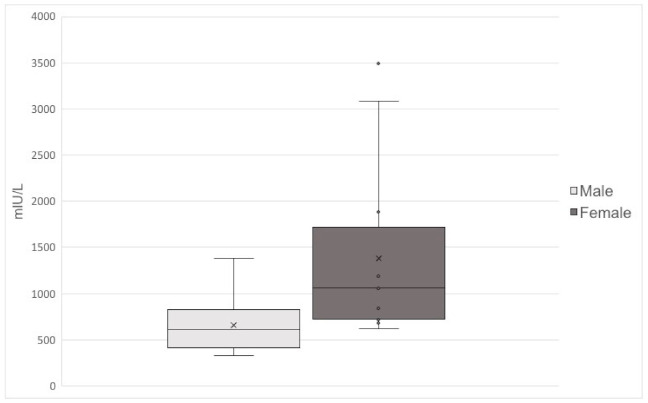
Prolactin levels in 33 men and 12 women with PWS and laboratory confirmed hyperprolactinemia.

**Figure 2 jcm-10-03613-f002:**
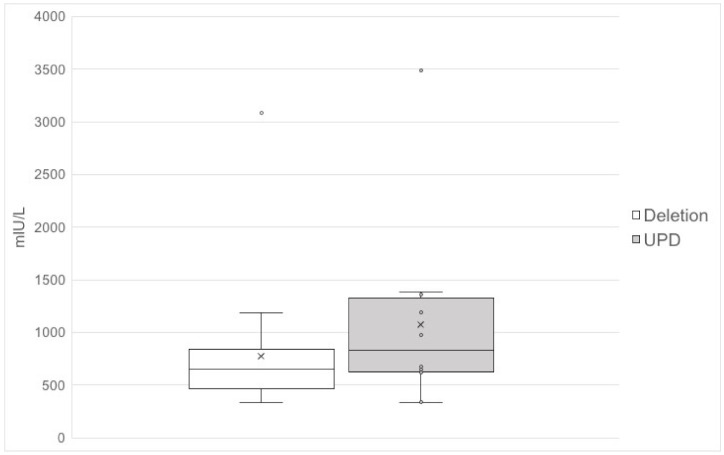
Prolactin levels in 24 adults with paternal chromosome 15q11–13 deletion and 14 adults with chromosome 15 maternal uniparental disomy.

**Table 1 jcm-10-03613-t001:** Characteristics of 45 adults with Prader-Willi syndrome and hyperprolactinemia. Results are shown as median and range, unless otherwise specified.

Characteristics of 45 Adults with Prader-Willi Syndrome and Hyperprolactinemia	
**Demographics**	N = 45
Men/women (*n*/*n*)	33/12
Age (years)	29 (19–58)
BMI (kg/m^2^)	29.8 (20.5–45.5)
Treatment with sex hormones (*n*)	29
Untreated hypogonadism (*n*)	16
Treatment with growth hormone (*n*)	24
Treatment with antipsychotics (*n*)	25
Treatment with serotonin re-uptake inhibitors (SSRI)	9
Pituitary microadenoma	1
**Hormone values**	
TSH (mU/L)	1.6 (0.3–3.8)
Free T4 (pmol/L)	15 (8.2–22.2)
Free T3 (pmol/L)	4.4 (2.9–5.4)
Prolactin (mIU/L)	680 (329–5950)

## Data Availability

The data are not publicly available due to privacy and ethical restrictions. The data that support the findings of this study are on request available from the corresponding author.

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
