# Peer review of "Hyperprolactinemia in Adults with Prader-Willi Syndrome"

_jcm, 2021, doi:10.3390/jcm10163613_

Round 1

Reviewer 1 Report

This is a multicenter study about a rare condition: PWS. A total of 441 clinical records of  PWS adults were reviewed. Although in a retrospective manner, I think results are innovative and interesting.

I have only minor questions

-Page 5 line 116-119: please rewrite this paraghraph. It’s difficult to understand

-Page 10, line 226: after samples there is a “?” that should be deleted

-Only 2 patients from 441 required cabergoline and only 1 patient had a microprolactinoma. Page 13 line 302. Don’t you think that it would be better to establish a higher cut off to recommend pituitary imaging? Having into account the challenge for performing MRI in PWS patients I think the profit it’s very low in cases with middle hyperprolactinemia.

Author Response

This is a multicenter study about a rare condition: PWS. A total of 441 clinical records of  PWS adults were reviewed. Although in a retrospective manner, I think results are innovative and interesting.

 I have only minor questions

 -Page 5 line 116-119: please rewrite this paragraph. It’s difficult to understand

Answer: Our apologies for the unclarity of the paragraph. It has now been rewritten.

-Page 10, line 226: after samples there is a “?” that should be deleted

Answer: Thank you for this notification. The question mark has been deleted.

-Only 2 patients from 441 required cabergoline and only 1 patient had a microprolactinoma. Page 13 line 302. Don’t you think that it would be better to establish a higher cut off to recommend pituitary imaging? Having into account the challenge for performing MRI in PWS patients I think the profit it’s very low in cases with middle hyperprolactinemia.

Answer: We thank the reviewer for this important comment. We agree with the reviewer’s arguments and have changed the cut off to >2200 mIU/L or >100 ng/L.  (Page 14, line 316-317).

Reviewer 2 Report

This manuscript investigates a demographic survey of hyperprolactinemia in individuals with Prader-Willi syndrome by using routine clinical assay based on retrospective international data in collaboration with several hospitals.

The aim is rational for better management of behavioral challenges. The hypothesis is clear. While the introduction is a slightly redundant and noncoherent, the methods and statistics are relevant. As a result, the findings are important to being aware of a potential side effect of antipsychotic medication. Below, I discuss some minor points that should be addressed if possible.

Introduction:

The authors described the significance of demographic survey of prolactin levels in individuals with PWS. However, how prolactin is likely to contribute to pathophysiology of PWS seems to be noncoherently introduced. Would it be possible to discuss these points in a more PWS-focused manner even if the prolactin might not have the central role in pathophysiology?

Discussion:

It may be a good idea to show previous reports on the demographics of hyperprolactinemia in other psychiatric or neurodevelopmental disorders to discuss whether the current data in PWS indicates any characteristics from a viewpoint of susceptibility to antipsychotic treatment. Moreover, it may be more advantageous to show the correlation analysis between dose of medication and prolactin levels, as it would be beneficial to be fully aware of possible risks for practitioners.

To conclude, the authors should reconsider these minor points to make the survey relevant and show how it can be novel and informative. I think the manuscript will be ready to publish after the minor corrections.

Author Response

This manuscript investigates a demographic survey of hyperprolactinemia in individuals with Prader-Willi syndrome by using routine clinical assay based on retrospective international data in collaboration with several hospitals.

The aim is rational for better management of behavioral challenges. The hypothesis is clear. While the introduction is a slightly redundant and noncoherent, the methods and statistics are relevant. As a result, the findings are important to being aware of a potential side effect of antipsychotic medication. Below, I discuss some minor points that should be addressed if possible.

Introduction:
The authors described the significance of demographic survey of prolactin levels in individuals with PWS. However, how prolactin is likely to contribute to pathophysiology of PWS seems to be noncoherently introduced. Would it be possible to discuss these points in a more PWS-focused manner even if the prolactin might not have the central role in pathophysiology?

Answer: Thank you for this comment. The Introduction has been rewritten and the text more focused on the adverse effects of hyperprolactinemia in PWS. 

Discussion:
It may be a good idea to show previous reports on the demographics of hyperprolactinemia in other psychiatric or neurodevelopmental disorders to discuss whether the current data in PWS indicates any characteristics from a viewpoint of susceptibility to antipsychotic treatment. Moreover, it may be more advantageous to show the correlation analysis between dose of medication and prolactin levels, as it would be beneficial to be fully aware of possible risks for practitioners.

Answer: We thank the reviewer for these interesting and important comments. Hyperprolactinaemia is a common adverse effect of antipsychotic drug therapy and we have no indications that patients with PWS have a greater susceptibility for developing hyperprolactinemia during anti-psychotic treatment than other patient groups. As 80% of our adults with PWS and hyperprolactinemia had a mild hyperprolactinemia and as certain antipsychotics can have profound effects on prolactin levels even in low doses, we did not calculate the correlation between doses and prolactin levels. We have added the following text and added a new reference [26]: “In comparison, cross-sectional studies of patients with schizophrenia and bipolar diseases have reported a prevalence of hyperprolactinemia of 44–75% in women and 23–72% in men [26]. Considering, that 80% of our PWS patients with hyperprolactinemia had only a mild hyperprolactinemia and that some antipsychotics (such as risperidone, paliperidone and amisulpride) might raise the serum prolactin level at relatively low doses [24], we have no indications that patients with PWS have a greater susceptibility for developing hyperprolactinemia during anti-psychotic treatment than other patient groups. (page 13-14, line 286-292).

To conclude, the authors should reconsider these minor points to make the survey relevant and show how it can be novel and informative. I think the manuscript will be ready to publish after the minor corrections.